# Modular Multimodal Alignment using Time-Series EHR Data for Enhancing Medical Image Classification

## Abstract

State-of-the-Art (SOTA) medical image classification models are generally pre-trained with large-scale data via self-supervised learning frameworks, to obtain high-quality image representations for radiology-based downstream tasks. The models are typically trained with a large amount of images or image-text pairs, where the text is extracted from the radiology report of the associated images. Such approaches neglect the rich contextual information available in the patient's Electronic Health Records (EHR), such as vital sign measurements and laboratory test results, which may be highly relevant for some modalities and tasks, such as those involving Chest X-Rays (CXR). Leveraging additional modalities during pre and/or post-training is not straightforward due to the small-scale of paired multimodal datasets. In this paper, we propose a new modular alignment strategy that leverages EHR data to enhance quality of representations of a pre-trained CXR image classification model, without requiring training from scratch. In particular, the framework employs a cross-modal learning objective to capture both global and localized interactions between CXR and EHR features. We ran experiments using the largest publicly-available multimodal dataset, specifically MIMIC-CXR and MIMIC-IV, to propose a new chest X-ray image classification model denoted as `MedCAM`. We evaluated `MedCAM` on several publicly available datasets. Our empirical findings show that it significantly outperforms a variety of SOTA baselines in terms of area under the receiver operating characteristic curve. The results highlight the benefit of leveraging EHR data and illustrate the potential of modular learning for efficient multimodal model enhancements.

## 1 Introduction

Chest X-Ray (CXR) image classification models, trained on large-scale chest radiograph datasets have achieved remarkable success in automated disease detection, localization, and radiology report generation (Paschali et al., 2025; Moor et al., 2023). These models have become a cornerstone in the development of Artificial Intelligence (AI) tools for clinical decision support, demonstrating strong generalization across institutions and diagnostic tasks. Most of the existing radiology models are primarily trained using unimodal or bimodal data (Paschali et al., 2025), specifically medical images and their associated free-text reports.

However, these approaches largely overlook important clinical information available in structured Electronic Health Records (EHR), such as vital signs and laboratory test results (Azad et al., 2023). The exclusion of EHR data limits the model's ability to contextualize imaging findings within the patient's broader clinical background, which is an essential component for accurate diagnosis and effective treatment planning (Paschali et al., 2025; Zhou et al.; Rajpurkar et al., 2022). For instance, a solitary pulmonary nodule identified on a routine CXR may appear suspicious using imaging alone, but its clinical interpretation heavily depends on contextual information in the EHR. Hence, we hypothesize that incorporating contextual clinical data during pre-training, typically documented in EHR, can enhance the quality of representations for image-based downstream prediction tasks.

Most existing research focuses on downstream fusion of EHR and CXR data (Hayat et al., 2022). Multimodal fusion approaches that combine imaging and clinical data have shown notable improve-

ments in diagnostic performance over unimodal baselines (Hayat et al., 2022; Yao et al., 2024). These findings are consistent with how radiologists process comprehensive EHR data to reduce diagnostic uncertainty, resulting in fewer follow-up imaging requests and more confident interpretations (Bowman, 2013; Castillo et al., 2021). A key challenge in multimodal fusion is the limited size of labeled paired datasets, which are substantially smaller than unimodal imaging corpora. This scarcity motivates our research question: *How can EHR data be effectively leveraged to enhance the performance of pretrained image-only classification models?* Recent efforts have begun to establish benchmarks for CXR–EHR pretraining (Elsharief et al., 2025). However, existing alignment strategies often fail to capture fine-grained interactions between specific imaging findings and their corresponding clinical variables. Moreover, pre-training from scratch using both modalities typically yields only modest gains, constrained by the limited availability of multimodal samples. Consequently, current approaches do not fully harness the complementary strengths of imaging and clinical data.

Given the aforementioned constraints, modular adaptation strategies offer a promising alternative by leveraging pretrained models, and enabling efficient cross-modal integration without full retraining. Moreover, while prior work has introduced a variety of multimodal architectures, most are evaluated on limited datasets and narrowly scoped tasks, making it difficult to assess their generalizability across clinically diverse scenarios. To address these challenges, we propose a new modular cross-modal alignment objective to achieve fine-grained relationships between imaging features and EHR context. We summarize our contributions as follows:

1. We propose a new modular cross-modal alignment objective that aligns imaging features with EHR context at both the global and local levels by leveraging a cross-attention mechanism, enabling the model to selectively extract and integrate the most relevant clinical information from the EHR data.

2. We empirically validate our approach on the largest publicly available multimodal dataset, MIMIC-CXR (Johnson et al., 2019) paired with MIMIC-IV (Johnson et al., 2021), and present `MedCAM`, a new chest X-ray classification model.

3. We conduct extensive experiments comparing `MedCAM` to state-of-the-art (SOTA) baselines, including image-only, image–EHR, and image-text pretraining strategies. Our results show that `MedCAM` achieves consistently strong and competitive performance across diverse clinical settings.

4. To support reproducibility, we make our code publicly available at `https://anonymous. 4open.science/r/foundation-cxr-D48B`.

## 2 RELATED WORK

**CXR Image Classification Models.** The emergence of AI models for CXR analysis has provided useful and transferable visual representations for various downstream medical tasks. We categorize CXR image-classification models according to the two most prominent pre-training approaches: (i) *image-only models* and (ii) *image-text models*. Image-only models leverage supervised learning or self-supervised learning approaches to extract meaningful representations from large collections of unlabeled CXR images. Models such as MoCo-CXR (Sowrirajan et al., 2021) and MGCA (Wang et al., 2022a) adopt contrastive learning frameworks in the radiology domain, while CheXzero (Tiu et al., 2022) employs a teacher-student framework to distill expert knowledge into model representations.

Image-text models jointly learn visual and textual representations by aligning CXR images with their corresponding radiology reports. Such models only leverage the reports during pre-training, as they are also often used to extract the groundtruth labels for classification tasks. ConVIRT (Zhang et al., 2022) pioneered this approach using a bidirectional contrastive loss between image and text embeddings, while CXR-BERT (Boecking et al., 2022) extends BERT with vision-language objectives specifically designed for radiology. Other models like GLoRIA (Huang et al., 2021) incorporate fine-grained image region-text alignment to capture localized abnormalities and their textual descriptions. Despite their impressive performance, existing CXR classification models exhibit two primary limitations. First, they do not incorporate EHR data, which contains essential contextual information such as patient history or laboratory results, that radiologists routinely consider during image assessment. Second, many existing models lack mechanisms for fine-grained integration

of specific clinical context with images, which is essential for enhancing diagnostic precision, particularly in subtle or complex conditions.

**Multimodal Self-Supervised Pre-Training.** Existing multimodal self-supervised pre-training strategies generally follow two broad paradigms: (i) *concurrent pre-training* and *modular pre-training*. In the concurrent pre-training paradigm, all modalities are jointly processed during training. Approaches such as MedViLL (Moon et al., 2022) and MedKLIP (Wu et al., 2023) leverage shared encoders or cross-modal attention to integrate multiple modalities simultaneously. REFERS (Zhou et al., 2022) further extends this idea by jointly optimizing image–text and image–EHR alignment through multi-head attention, capturing rich inter-modality relationships. Similarly, MeTra (Khader et al., 2023) adopts a transformer-based architecture to integrate CXR images with clinical variables for ICU mortality prediction.

In contrast, the modular pre-training paradigm decouples unimodal and multimodal learning. Unimodal encoders are first trained independently and later adapted for multimodal integration. For example, LMFusion (Shi et al., 2024) extends pretrained language models with visual features through efficient adaptation, while Zhou et al. (2022) demonstrate that parameter-efficient adaptation techniques can achieve competitive performance in medical vision–language tasks at a fraction of the cost of full pre-training. Notably, most existing CXR-EHR models adopt concurrent pre-training, requiring costly end-to-end training from scratch (Elsharief et al., 2025). In contrast, modular adaptation, which is widely explored in language and vision-language models, remains underexplored for radiology foundation models with structured EHR data.

This gap highlights the need for efficient modular strategies tailored to medical imaging and clinical time-series integration. In this study, we focus our scope on modular image–EHR alignment, as opposed to image–text alignment, for two key reasons. First, aligning images with structured EHR data is inherently more challenging than image–text alignment, since radiology reports often encode downstream task information (e.g., diagnoses or impressions). Second, EHR data introduces greater heterogeneity and sparsity compared to free-text reports, making modular alignment a more rigorous test of cross-modal integration. Additionally, we restrict our evaluation to image-only classification downstream tasks, leaving multimodal fusion approaches outside the scope of this study.

## 3 METHODOLOGY

We propose a modular framework designed to integrate structured clinical data with pre-trained CXR foundation models. The objective is to enable clinically-aware adaptation of vision models without requiring full retraining, by aligning latent representations from EHR and imaging data. We describe our methodology in three main sections: (1) preliminaries and task formulation (Section 3.1), (2) modality-specific feature encoding (Section 3.2), and (3) the clinical adaptation mechanism and training objectives (Section 3.3).

### 3.1 PRELIMINARIES

Let $\mathbf{x}_{\text{CXR}} \in \mathbb{R}^{w \times h}$ denote a single CXR image of size $w \times h$ for a given patient, and let $\mathbf{x}_{\text{EHR}} \in \mathbb{R}^{d \times t}$ represent the associated multivariate time-series EHR data, where $d$ is the number of clinical features and $t$ is the number of timesteps. Given a pre-trained CXR foundation model $f_{\text{CXR}}$, the goal is to adapt it to produce predictions $\hat{y}$ for downstream tasks. The adapted model, denoted as $\tilde{f}_{\text{CXR}}$, is enhanced through integration with EHR data.

### 3.2 FEATURE ENCODING

First, we leverage modality-specific encoders to embed both CXR and EHR data into a shared latent space. For the CXR modality, we use the pre-trained vision encoder $f_{\text{CXR}}$ to extract a feature representation:

$$\mathbf{z}_{\text{CXR}} = f_{\text{CXR}}(\mathbf{x}_{\text{CXR}}). \tag{1}$$

For EHR data, we define a temporal encoder $f_{\text{EHR}}$, typically a neural network tailored for time-series data, to obtain a corresponding latent representation:

$$\mathbf{z}_{\text{EHR}} = f_{\text{EHR}}(\mathbf{x}_{\text{EHR}}). \tag{2}$$

These representations serve as the inputs of the multimodal clinical-adapted pre-training strategy below.

## 3.3 ADAPTATION MODULE

The adaptation module is the core component of our approach, enabling the pre-trained CXR model to incorporate EHR data efficiently. We assume the presence of an *anchor modality* that serves as the primary representation space for alignment. The choice of anchor modality is guided by domain knowledge and informed by clinical guidelines and task-specific literature. For a given task domain—defined as a set of related downstream tasks, we identify the modality that provides the most critical diagnostic or predictive signal. In our setting, this is the CXR modality, which we designate as the anchor. Consequently, EHR representations are adapted to align with the CXR latent space.

### 3.3.1 MULTI-LEVEL ALIGNMENT

We introduce multimodal alignment by projecting CXR and EHR representations into a shared semantic space. To capture both high-level and fine-grained relationships, we implement two complementary strategies: (i) global-level alignment and (ii) local-level alignment, both trained using bidirectional contrastive learning.

**Global-Level Cross-Modal Alignment.** Global alignment ensures that holistic embeddings from the CXR and EHR modalities are compatible in a unified latent space. Let $\mathbf{z}_{\text{CXR}}$ and $\mathbf{z}_{\text{EHR}}$ denote the global representations of the CXR image and EHR data, respectively. We optimize a symmetric contrastive objective over paired samples in a batch of size $B$:

$$\mathcal{L}_{\text{CXR}\rightarrow\text{EHR}} = -\frac{1}{B} \sum_{i=1}^{B} \log \frac{\exp\left(\text{sim}(\mathbf{z}_{\text{CXR},i}, \mathbf{z}_{\text{EHR},i})/\tau\right)}{\sum_{j=1}^{B} \exp\left(\text{sim}(\mathbf{z}_{\text{CXR},i}, \mathbf{z}_{\text{EHR},j})/\tau\right)}, \quad (3)$$

$$\mathcal{L}_{\text{EHR}\rightarrow\text{CXR}} = -\frac{1}{B} \sum_{i=1}^{B} \log \frac{\exp\left(\text{sim}(\mathbf{z}_{\text{EHR},i}, \mathbf{z}_{\text{CXR},i})/\tau\right)}{\sum_{j=1}^{B} \exp\left(\text{sim}(\mathbf{z}_{\text{EHR},i}, \mathbf{z}_{\text{CXR},j})/\tau\right)}, \quad (4)$$

where $\text{sim}(\cdot, \cdot)$ denotes cosine similarity, and $\tau$ is a temperature scaling parameter. The overall global alignment loss is given by:

$$\mathcal{L}_{\text{global}} = \frac{1}{2} \left(\mathcal{L}_{\text{CXR}\rightarrow\text{EHR}} + \mathcal{L}_{\text{EHR}\rightarrow\text{CXR}}\right). \quad (5)$$

This contrastive loss encourages positive pairs (CXR–EHR from the same patient) to have higher similarity, while distinguishing them from mismatched pairs in the batch. Bidirectional alignment promotes mutual consistency across modalities.

**Local-Level Alignment using Top-$k$ Retrieval.** To enhance the granularity of cross-modal understanding, the local-level alignment component models fine-grained associations between CXR images and individual clinical features from EHR data. To capture fine-grained cross-modal dependencies, we define a local alignment objective based on the attention between EHR timesteps and the global CXR embedding. Let $A \in \mathbb{R}^{B \times t}$ denote the attention weights (after softmax) for a batch of size $B$, where $t$ is the number of EHR timesteps. For each CXR image, we select the top-$k$ EHR timesteps with the highest attention scores relative to the CXR representation, where $k$ is a hyperparameter. Let $z_{\text{EHR},j}$ denote the embedding of a selected EHR timestep, and $z_{\text{CXR},b}$ the global CXR embedding for the $b$-th patient. The bidirectional local alignment losses are defined as:

Table 1: **Summary of evaluation datasets.** We report the data splits (train/validation/test) for the in-distribution dataset (MIMIC-CXR), which is paired with EHR data during pretraining, and for three out-of-distribution datasets (NIH, VinDr-CXR, CheXpert) that lack EHR information and are therefore unseen during multimodal alignment. This setup allows us to assess both in-distribution performance and cross-dataset generalization.

|  | Dataset | Train / Val / Test (%) |
|---|---|---|
| *In-distribution* | MIMIC-CXR (Johnson et al., 2019) | 60 / 20 / 20 |
| *Out-of-distribution* | NIH (Wang et al., 2017) | 70 / 10 / 20 |
|  | VinDr-CXR (Nguyen et al., 2022) | 67 / 16 / 17 |
|  | CheXpert (Irvin et al., 2019) | 99.6 / 0.3 / 0.1 |

$$\mathcal{L}_{\text{CXR}\rightarrow\text{EHR}}^{\text{local}} = -\frac{1}{B \cdot k} \sum_{b=1}^{B} \sum_{j \in \text{Top-}k(b)} \log \frac{\exp\left(\text{sim}(z_{\text{CXR},b}, z_{\text{EHR},j})/\tau\right)}{\sum_{j'=1}^{t} \exp\left(\text{sim}(z_{\text{CXR},b}, z_{\text{EHR},j'})/\tau\right)}, \tag{6a}$$

$$\mathcal{L}_{\text{EHR}\rightarrow\text{CXR}}^{\text{local}} = -\frac{1}{B \cdot k} \sum_{b=1}^{B} \sum_{j \in \text{Top-}k(b)} \log \frac{\exp\left(\text{sim}(z_{\text{EHR},j}, z_{\text{CXR},b})/\tau\right)}{\sum_{b'=1}^{B} \exp\left(\text{sim}(z_{\text{EHR},j}, z_{\text{CXR},b'})/\tau\right)}. \tag{6b}$$

The final local alignment loss is obtained by averaging both directions:

$$\mathcal{L}_{\text{local}} = \tfrac{1}{2}\left(\mathcal{L}_{\text{CXR}\rightarrow\text{EHR}}^{\text{local}} + \mathcal{L}_{\text{EHR}\rightarrow\text{CXR}}^{\text{local}}\right). \tag{7}$$

This bidirectional formulation ensures that only the most informative EHR timesteps are consistently aligned with the CXR representations.

**Overall Training Objective.** The total loss for obtaining the adapted $\tilde{f}_{\text{CXR}}$ using $\texttt{MedCAM}$ combines both global and local alignment objectives:

$$\mathcal{L}_{\text{overall}} = \mathcal{L}_{\text{global}} + \mathcal{L}_{\text{local}} \tag{8}$$

This joint objective enables the model to capture both high-level consistency and fine-grained cross-modal dependencies between the CXR images and clinical data.

### 3.3.2 FINE-TUNING STEP

Following multimodal alignment, the adapted encoder $\tilde{f}_{\text{CXR}}$ is fine-tuned for specific downstream tasks using supervised learning. Let $y$ denote the ground truth label for a given prediction task. We employ the Binary Cross-Entropy (BCE) loss to optimize the model outputs:

$$\mathcal{L}_{\text{FT}} = \text{BCE}\left(y, \tilde{f}_{\text{CXR}}(\mathbf{x}_{\text{CXR}})\right), \tag{9}$$

where $\tilde{f}_{\text{CXR}}(\mathbf{x}_{\text{CXR}})$ represents the task-specific prediction based on the adapted CXR encoder.

## 4 EXPERIMENTS

### 4.1 CLINICAL DATASETS AND TASK

To conduct our experiments, we used the MIMIC-CXR dataset Johnson et al. (2019) and associated EHR data in the MIMIC-IV dataset Johnson et al. (2021). The EHR data extracted from MIMIC-IV includes both categorical and continuous features. These features encompass vital signs, laboratory measurements, and clinical assessment scores. We implement a three-stage preprocessing pipeline

for the EHR data: (1) temporal resampling of irregularly collected features to a fixed sampling rate ($\Delta t = 2$ hours), (2) missing value imputation using the most recent available measurement or normal reference values when no prior measurements exist, and (3) data transformation by one-hot encoding all categorical variables. This resulted with a data matrix consisting of 76 features encapsulating both categorical and continuous features, such that $x_{\text{EHR}} \in \mathbb{R}^{d \times t}$, where $d = 76$ and $t$ is the number of time-steps. We also standardize the continuous variables to ensure consistency across different scales. To process the CXR images, we followed the same pre-processing steps as in previous work Zhou et al. (2024), including image resizing to 256×256, random cropping to 224×224, and normalizing. We parameterized $f_{\text{CXR}}$ as a pre-trained ResNet-50 encoder adopted from BarlowTwins-CXR (Sheng et al., 2024), and $f_{\text{EHR}}$ as an LSTM (Hayat et al., 2022). The global alignment loss considers the fully encoded EHR sequence, while in the local alignment loss and we set $k = 10$ in the adaptation step based on empirical findings.

We then evaluated our adapted encoder, denoted as `MedCAM`, by fine-tuning it on multi-label image classification across the in-distribution MIMIC-CXR test set and out-of-distribution test sets, i.e. datasets that were not seen during pre-training since they are not associated with any EHR data. We used widely adopted external datasets including MIMIC-CXR, CheXpert, and VinDR-CXR, with a summary provided in Table 1. Additional information on the datasets can be found in the Supplementary Information. For consistency, we follow standardized preprocessing pipelines, training, validation and test splits, and evaluation protocols from prior work (Zhou et al., 2024; Yu et al., 2023). We report performance on the test set using macro-averaged Area under the Receiver Operating Characteristic curve (AUROC) and Area under the Precision Recall Curve (AUPRC) as the main metrics.

## 4.2 BASELINE MODELS

We compare the performance of our model against several representative baselines. We organize these baselines into two primary groups: image-only models and image-EHR pretrained models. This division highlights the differences in training paradigms and data modalities leveraged by each approach. We also conducted an additional analysis to compare the performance of our model to image-text pre-trained models, although they are not directly comparable since the radiology reports are associated with the downstream classification labels.

**Image-Only Models.** Baselines relying on CXR images only for pretraining:

- *Supervised Learning (ResNet-50)*: a standard supervised training baseline, serving as a reference for conventional fully supervised pipelines (He et al., 2016).
- *BarlowTwins-CXR (ResNet-50)*: applies a self-supervised Barlow Twins framework for CXR image encoding to improve abnormality localization and address domain inconsistency in cross-dataset settings. The model uses the NIH dataset during self-supervised pre-training followed by supervised fine-tuning on VinDr-CXR (Sheng et al., 2024).
- *EVA-X (Dual ViT)*: introduces a scalable, self-supervised foundation model for CXR analysis that integrates contrastive learning and masked image modeling for pre-training without human annotations (Yao et al., 2025). We evaluate the Tiny (T) and Small (S) versions in our experiments.

**Image-EHR Models.** Baselines that jointly leverage CXR images and paired EHR data to learn multimodal representations (Elsharief et al., 2025):

- *ConVIRT*: pretrains medical image encoders using a bidirectional contrastive learning objective between CXR images and their paired EHR data (Zhang et al., 2022; Elsharief et al., 2025).
- *VICReg*: a variance-invariance-covariance regularization approach that learns joint multimodal embeddings by encouraging similarity across paired samples while maintaining feature diversity and reducing redundancy (Bardes et al., 2022; Elsharief et al., 2025).
- *ALIGN*: aligns multi-modal embeddings using a contrastive loss (Jia et al., 2021; Elsharief et al., 2025).

Table 2: **In-distribution evaluation of multi-label image classification performance on the MIMIC-CXR dataset.** Comparison of supervised learning, image-EHR pretraining, image-only pretraining, and our proposed image-EHR adaptation `MedCAM`. (**Best**, Second Best, 95% CI).

| Pretraining Strategy | Model | AUROC | AUPRC |
|---|---|---|---|
| *Supervised* | ResNet-50 | $0.705_{\pm 0.013}$ | $\mathbf{0.323}_{\pm 0.006}$ |
| *Image-EHR* | ConVIRT | $0.638_{\pm 0.016}$ | $0.259_{\pm 0.018}$ |
| | VICReg | $0.615_{\pm 0.017}$ | $0.261_{\pm 0.017}$ |
| | ALIGN | $0.595_{\pm 0.018}$ | $0.233_{\pm 0.018}$ |
| *Image-Only* | BarlowTwins-CXR | $\underline{0.809}_{\pm 0.003}$ | $\underline{0.287}_{\pm 0.007}$ |
| | EVA-X-T | $0.791_{\pm 0.003}$ | $0.242_{\pm 0.022}$ |
| | EVA-X-S | $0.799_{\pm 0.003}$ | $0.273_{\pm 0.016}$ |
| *Image-EHR Adaptation* | `MedCAM` (Ours) | $\mathbf{0.906}_{\pm 0.003}$ | $0.244_{\pm 0.013}$ |

**Image-Text Models.** Baselines that jointly leverage CXR images and paired radiology reports data to learn multimodal representations.

- *ConVIRT (ResNet-50)*: pretrains medical image encoders using a bidirectional contrastive learning objective between CXR images and their paired radiology reports, eliminating the need for manual labels or rule-based extraction (Zhang et al., 2022).

- *GLoRIA (ResNet-50)*: introduces an attention-based multimodal framework that learns both global and local alignments between image regions and radiology report text via contrastive learning (Huang et al., 2021).

- *MedCLIP (ResNet-50/Swin ViT)*: enhances contrastive vision-language pretraining for medical images by decoupling image-text pairs and introducing a semantic matching loss that mitigates false negatives common in clinical data (Wang et al., 2022b).

### 4.3 IMPLEMENTATION DETAILS

During fine-tuning, we conduct a grid search to vary the learning rate hyperparameter across the following search space of values: $\{1 \times 10^{-2}, 1 \times 10^{-3}, 1 \times 10^{-5}\}$. For multi-label classification, we follow the approach in previous work (Zhou et al., 2024) and append a linear classifier to a pre-trained image encoder. We fine-tune both the encoder and classifier head on each dataset using the binary cross entropy loss for 200 epochs with a training batch size of 64. The multi-label AUROC (macroaveraged) on the validation set is monitored every 5 epochs to assess whether the model may be overfitting to activate early stopping. After fine-tuning, the weights of the encoder and classifier corresponding to the highest AUROC score on the validation set are used to generate the multi-label AUROC on the test set. We compute 95% confidence intervals using bootstrapping.

## 5 RESULTS

We evaluate our proposed model `MedCAM` on the multi-label chest X-ray classification task, comparing against supervised, image-only, and image-text pretraining baselines. Results are organized into three parts: in-distribution evaluation on the MIMIC-CXR dataset, generalization to out-of-distribution datasets (NIH, VinDr-CXR, CheXpert), and direct comparison with state-of-the-art image-text pretraining methods.

### 5.1 EVALUATION ON THE MIMIC-CXR DATASET

Table 2 reports the in-distribution performance of all baselines on MIMIC-CXR. Among image-only models, *BarlowTwins-CXR* achieves the highest AUROC (0.809 ± 0.003), outperforming *EVA-X* variants. However, our proposed `MedCAM` substantially surpasses all baselines, achieving 0.906 ± 0.003, a relative improvement of nearly 10% over the strongest image-only baseline. This demonstrates that incorporating structured EHR information during pretraining provides substantial performance gains evaluated across different strategies on the same data distribution.

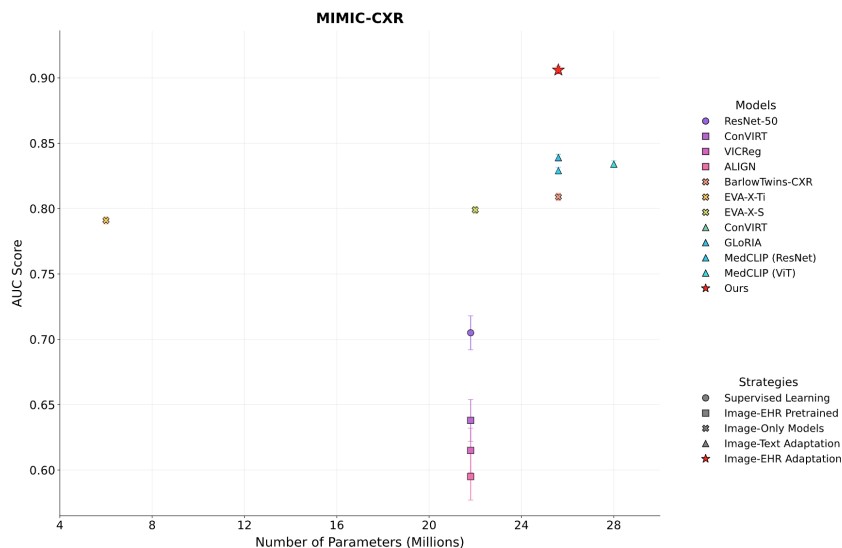

Figure 1: Comparison of different models and pre-training strategies using the MIMIC-CXR dataset.

Table 3: **Out-of-distribution evaluation of multi-label image classification performance.** We report AUROC (95% CI) on three external chest X-ray benchmarks: NIH, VinDr-CXR, and CheXpert. Results compare image-only pretraining approaches with our proposed image-EHR adaptation method (`MedCAM`). (**Best**, Second Best, 95% CI)

| Pretraining Strategy | Model | NIH | VinDr-CXR | CheXpert |
|---|---|---|---|---|
| *Image-Only* | BarlowTwins-CXR | 0.778 $_{\pm 0.004}$ | 0.841 $_{\pm 0.012}$ | 0.861 $_{\pm 0.022}$ |
| | EVA-X-T | 0.736 $_{\pm 0.004}$ | 0.690 $_{\pm 0.022}$ | 0.836 $_{\pm 0.021}$ |
| | EVA-X-S | 0.745 $_{\pm 0.005}$ | 0.728 $_{\pm 0.021}$ | 0.845 $_{\pm 0.017}$ |
| *Image-EHR Adaptation* | `MedCAM` (Ours) | **0.847** $_{\pm 0.004}$ | **0.852** $_{\pm 0.016}$ | **0.872** $_{\pm 0.024}$ |

## 5.2 Evaluation on Out-of-Distribution Datasets

Table 3 summarizes model generalization to NIH, VinDr-CXR, and CheXpert datasets. Among image-only models, *BarlowTwins-CXR* consistently ranks second across all datasets, highlighting the strength of self-supervised image-only pretraining. `MedCAM` achieves the best performance on all three benchmarks (NIH: 0.847 ± 0.004, VinDr: 0.852 ± 0.016, CheXpert: 0.872 ± 0.024), demonstrating robust cross-dataset generalization. These results indicate that modular pre-training, where a pre-trained encoder is adapted with an additional modality, significantly enhances the performance on downstream tasks compared to concurrent pre-training with multiple modalities.

## 5.3 Comparison with Image-Text Models

Finally, Table 4 compares `MedCAM` with state-of-the-art image-text pretrained models. While GLoRIA achieves the highest AUROC on VinDr-CXR (0.905 ± 0.008), and MedCLIP-ViT performs best on CheXpert (0.879 ± 0.022), `MedCAM` achieves the best results on MIMIC-CXR (0.906 ± 0.003) and NIH (0.847 ± 0.004), with comparable performance on CheXpert (0.872 ± 0.024). Overall, `MedCAM` delivers competitive or superior performance across datasets without requiring paired image-text data during pretraining, highlighting the effectiveness of modular adaptation that leverages EHR data.

## 6 Discussion & Conclusion

This work presents a modular framework that enhances existing CXR foundation models by integrating clinical information in the EHR data. One key strength of our proposed approach is that it

Table 4: **In-distribution and out-of-distribution performance across pretraining strategies.** We evaluate AUROC (95% CI) on the MIMIC-CXR dataset (in-distribution) and three external benchmarks (NIH, VinDr-CXR, CheXpert). Results include contrastive image-text models (ConVIRT, GLoRIA, MedCLIP variants) and our proposed image-EHR adaptation method `MedCAM`. (**Best**, Second Best

| Pretraining Strategy | Model | MIMIC-CXR | NIH | VinDr | CheXpert |
|---|---|---|---|---|---|
| *Image-Text* | ConVIRT | 0.839 $_{\pm 0.003}$ | 0.819 $_{\pm 0.004}$ | 0.904 $_{\pm 0.008}$ | 0.873 $_{\pm 0.023}$ |
| | GLoRIA | 0.839 $_{\pm 0.003}$ | 0.814 $_{\pm 0.004}$ | **0.905** $_{\pm 0.008}$ | 0.877 $_{\pm 0.021}$ |
| | MedCLIP-ResNet | 0.829 $_{\pm 0.003}$ | 0.817 $_{\pm 0.005}$ | 0.836 $_{\pm 0.014}$ | 0.861 $_{\pm 0.024}$ |
| | MedCLIP-ViT | 0.834 $_{\pm 0.003}$ | 0.820 $_{\pm 0.005}$ | 0.900 $_{\pm 0.009}$ | **0.879** $_{\pm 0.022}$ |
| *Image-EHR Adaptation* | `MedCAM` (Ours) | **0.906** $_{\pm 0.003}$ | **0.847** $_{\pm 0.004}$ | 0.852 $_{\pm 0.016}$ | 0.872 $_{\pm 0.024}$ |

does not require pretraining from scratch or access to large-scale paired multimodal datasets, which addresses a major scalability bottleneck in clinical multimodal learning. The main strength lies in the use of multi-level alignment objectives to learn both global and fine-grained interactions between imaging and clinical features. This design is motivated by how radiologists incorporate both holistic impressions and specific contextual cues when interpreting imaging findings. Our use of bidirectional contrastive learning further ensures mutual consistency between modalities. In terms of empirical validation, `MedCAM` demonstrates consistent performance gains across different public datasets and multiple downstream tasks. The method achieves results comparable to full multimodal models while maintaining higher flexibility and lower training cost. These results support the generalizability of the proposed framework and highlight the benefit of modular adaptation for efficient multimodal integration in clinical settings.

While MedCAM provides an efficient and extensible framework for incorporating EHR information into CXR foundation models, there are several limitations that could be addressed in the future research. First, the current design focuses on contrastive alignment at both global and local levels. Although this enables coarse and fine-grained cross-modal consistency, the alignment objective does not model higher-order interactions among clinical variables. More advanced alignment strategies, such as graph-based contrastive objectives or masked prediction tasks across modalities, may further enhance the model's capacity to capture clinically meaningful relationships.

Second, we only consider EHR data as the auxiliary modality. In practice, CXR studies are often accompanied by radiology reports, which contain detailed textual descriptions of imaging findings and clinical impressions. These reports provide complementary semantic cues that are not present in structured EHR fields, as evident by the performance of image-text models. However, such reports are used to extract the downstream classification labels, hence we focused on integrating EHR only. Extending MedCAM to integrate both EHR and textual reports as additional input modalities may yield more comprehensive multimodal representations, particularly for other tasks, like report generation or fine-grained phenotype classification.

Lastly, although our experiments include multiple publicly available datasets, they remain limited in size and scope compared to real-world clinical populations. Larger-scale evaluation across more diverse patient cohorts and healthcare systems are needed to fully assess the generalizability and robustness of MedCAM.

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
