# OpenReview forum: "Modular Multimodal Alignment using Time-Series EHR Data for Enhancing Medical Image Classification"
_ICLR.cc/2026/Conference — ICLR 2026 Conference Withdrawn Submission_

### Official Review · Reviewer_bSie · 2025-10-30

**Soundness:** 3
**Presentation:** 3
**Contribution:** 2
**Rating:** 4
**Confidence:** 4

**Summary:**

This paper introduces a framework to align x-ray data with EHR. Each modality has their own specific encoder, then a global + local alignment method is applied. Then finally the model is fine tuned on targeted tasks. It tests on datasets without EHR as out of distribution eval

**Strengths:**

- EHR alignment for medical image pretrianing is few so the motivation is strong
- On top of accuracy, the authors pay attention to computational cost, practicality for clinical deployment, using existing models effectively etc.

**Weaknesses:**

- the paper never rigorously enforces causal ordering between EHR and CXR timestamps. The temporal scope of EHR inputs is vaguely defined (“resampled at 2-hour intervals”) without specifying whether future observations relative to the imaging timestamp were excluded. In MIMIC-IV, many physiological measurements (e.g., labs, vitals) are backfilled retrospectively — meaning the EHR embedding could inadvertently encode post-image outcomes such as interventions or ICU status changes. This introduces temporal leakage, undermining the causal interpretability of the learned representations.
- It's still somewhat unclear how does this alignment actually help by conducting ablation study, e.g. via cross-modal retrieval accuracy, embedding similarity metrics, or attention interpretability. Further, the authors claim the adaptation captures “fine-grained interactions,” yet no visualization or ablation supports this. There’s also no control experiment comparing random EHR pairs or shuffled time windows — so it’s unclear whether the model is learning true cross-modal structure or merely benefiting from regularization.
- The theory is not systematically studied wrt why it works but merely empirically results. Dual contrastive loss (global/local) is not theoretically innovative. So further negative control study or failure case analysis is missing where something like shuffling EHR–CXR pairs, or perturbing EHR sequences temporally can help to understand more about the method

**Questions:**

- For EHR, how do you ensure that EHR sequences used for alignment include only data prior to or concurrent with the CXR acquisition time
- Any quantitative evidence that global and local alignment help on what aspect individually

---

> ### Author Response · Authors · 2025-12-03
>
> Dear Reviewer bSie,
>
> Thank you for your review and feedback.
>
> > The paper never rigorously enforces causal ordering between EHR and CXR timestamps.
>
> >The temporal scope of EHR inputs is vaguely defined (“resampled at 2-hour intervals”) without specifying whether future observations relative to the imaging timestamp were excluded. In MIMIC-IV, many physiological measurements (e.g., labs, vitals) are backfilled retrospectively — meaning the EHR embedding could inadvertently encode post-image outcomes such as interventions or ICU status changes. This introduces temporal leakage, undermining the causal interpretability of the learned representations.
> > For EHR, how do you ensure that EHR sequences used for alignment include only data prior to or concurrent with the CXR acquisition time
>
> We appreciate your concern regarding temporal leakage. To  construct our multimodal dataset from MIMIC-IV and MIMIC-CXR, we included paired samples where both the CXR and EHR modalities are present in a given ICU stay, following previous work such as in MedFuse. This means that the EHR data is resampled at 2-hour intervals throughout the full ICU stay. However, we understand the concern and will consider alignment for each CXR image with only EHR embeddings taken before the CXR acquisition time in future iterations of the work.
>
> > It's still somewhat unclear how does this alignment actually help by conducting ablation study, e.g. via cross-modal retrieval accuracy, embedding similarity metrics, or attention interpretability. Further, the authors claim the adaptation captures “fine-grained interactions,” yet no visualization or ablation supports this. There’s also no control experiment comparing random EHR pairs or shuffled time windows — so it’s unclear whether the model is learning true cross-modal structure or merely benefiting from regularization.
>
> > The theory is not systematically studied wrt why it works but merely empirically results. Dual contrastive loss (global/local) is not theoretically innovative. So further negative control study or failure case analysis is missing where something like shuffling EHR–CXR pairs, or perturbing EHR sequences temporally can help to understand more about the method
>
> > Any quantitative evidence that global and local alignment help on what aspect individually
>
> We appreciate the reviewer's suggestions and acknowledge the need for a more comprehensive experimental evaluation.  We have conducted an ablation study to compare the individual contributions of the global and local alignment losses. The results of this experiment on the MIMIC-CXR dataset are presented below:
>
> | Model | AUROC | AUPRC |
> |-------|-------|-------|
> | Supervised baseline (ResNet-50) | 0.705 | 0.323 |
> | MedCAM (global loss only) | 0.882 | 0.221 |
> | MedCAM (global + local loss) | 0.906 | 0.244 |
>
>
> These results quantitatively show that both the global and local losses contribute to the overall model performance (with a larger contribution from the global loss, as expected). The combination of both losses achieved the optimal performance.
>
> Moreover, we agree with the reviewer that a control experiment where the EHR-CXR pairs are shuffled will contribute to understanding the performance of the model and demonstrating that the model is learning from the cross-modal signal. We also acknowledge the need for further analysis such as attention interpretability as well as visualizations. We will ensure to incorporate the reviewer’s suggestions in future iterations of the work.

---

### Official Review · Reviewer_ZPP3 · 2025-10-31

**Soundness:** 2
**Presentation:** 2
**Contribution:** 2
**Rating:** 2
**Confidence:** 5

**Summary:**

The paper extends existing medical CLIP approaches—originally trained on image–text pairs—to image–EHR pairs, investigating how replacing textual reports with EHR data as a semi-supervised signal affects the performance of chest X-ray image encoders. However, the methodology lacks significant novelty beyond the dataset modification, the experimental evaluation is incomplete, and the writing and formatting are unclear.

**Strengths:**

1. The paper provides a thoughtful analysis of prior medical CLIP methods that rely on image–text pairs and articulates the potential advantages of incorporating EHR information during training.
2. It integrates the MIMIC-CXR and MIMIC-IV datasets to construct a new multimodal benchmark.

**Weaknesses:**

**I. Writing and Presentation**

1. The manuscript includes no figure illustrating the model architecture or overall framework.
2. The experimental section is poorly organized. Section 5.1 is titled “Evaluation on the MIMIC-CXR Dataset,” yet Table 2 does not include results for image–text baselines on MIMIC-CXR; those results are instead presented in Table 4. This disjointed presentation hinders direct comparison and impairs readability.
3. Figure 1 is the only figure in the paper, which is of low quality, difficult to interpret, and conveys minimal information.

**II. Methodology and Experiments**

1. Aside from constructing a new dataset and substituting EHR for textual reports, the paper introduces no clear methodological innovation. The combination of global and local alignment strategies has already been thoroughly explored in prior works such as MGCA[1], GLORIA[2], and Prior[3].
2. The comparison with image–text models is methodologically inconsistent. The visual encoders of the baseline image–text models are trained from scratch, whereas MedCAM employs a pre-trained encoder. This discrepancy likely explains the observed performance gap: MedCAM significantly outperforms baselines on MIMIC-CXR and NIH but underperforms on CheXpert and VinDr.
3. The selected image–text baselines are outdated. The most recent cited work is MedCLIP (EMNLP 2022). At a minimum, the paper should include 2–3 stronger, more recent baselines from 2024–2025.
4. Recent studies such as MedBIND [4] and M3Bind [5] have already explored multimodal alignment involving more than two modalities (e.g., image, text, and EHR simultaneously). The choice to replace text with EHR—rather than integrating all available modalities—requires justification.
5. The paper lacks any form of qualitative or representational visualization (e.g., t-SNE plots), which would help assess the quality of learned embeddings.

**Questions:**

1. Reorganize the manuscript’s writing and layout, particularly the design and placement of figures and tables, to enhance clarity and facilitate comparison.
2. Clearly articulate the novelty of the proposed approach beyond dataset construction.
3. Address the concern raised in Weakness II.2 regarding the unfair comparison due to differing training protocols (from-scratch vs. pre-trained encoders).
4. Include more recent and competitive baselines (e.g., BiomedCLIP, M3Bind, or other 2024–2025 medical multimodal models) in the experimental comparison.
5. Justify the design choice of replacing textual reports with EHR instead of jointly leveraging image, text, and EHR in a unified framework.
6. Add visualization experiments (e.g., t-SNE, attention maps, or embedding space analyses) to support the claims and improve interpretability.

[1] Wang, F., Zhou, Y., Wang, S., Vardhanabhuti, V., & Yu, L. (2022). Multi-granularity cross-modal alignment for generalized medical visual representation learning. *Advances in neural information processing systems*, *35*, 33536-33549.
[2] Huang, S. C., Shen, L., Lungren, M. P., & Yeung, S. (2021). Gloria: A multimodal global-local representation learning framework for label-efficient medical image recognition. In *Proceedings of the IEEE/CVF international conference on computer vision* (pp. 3942-3951).
[3] Cheng, P., Lin, L., Lyu, J., Huang, Y., Luo, W., & Tang, X. (2023). Prior: Prototype representation joint learning from medical images and reports. In *Proceedings of the IEEE/CVF international conference on computer vision* (pp. 21361-21371).
[4]Gao, Y., Kim, S., Austin, D. E., & McIntosh, C. (2024, October). MEDBind: Unifying Language and Multimodal Medical Data Embeddings. In *International Conference on Medical Image Computing and Computer-Assisted Intervention* (pp. 218-228). Cham: Springer Nature Switzerland.
[5]Liu, Y., Xi, S., Liu, S., Ding, H., Jin, C., Zhong, C., ... & Shen, Y. (2025). Multimodal Medical Image Binding via Shared Text Embeddings. *arXiv preprint arXiv:2506.18072*.

---

> ### Author Response · Authors · 2025-12-03
>
> Dear Reviewer ZPP3,
>
>  Thank you for your thorough review and constructive feedback.
>
> > The manuscript includes no figure illustrating the model architecture or overall framework.
>
> > Figure 1 is the only figure in the paper, which is of low quality, difficult to interpret, and conveys minimal information.
>
> We appreciate the reviewer’s feedback regarding the figures. We agree that including a main figure illustrating the model architecture would enhance the clarity of the paper. We will ensure that we incorporate this figure in the revised manuscript, and will work on improving the quality of the existing Figure 1.
>
> > The experimental section is poorly organized. Section 5.1 is titled “Evaluation on the MIMIC-CXR Dataset,”...
>
> > Reorganize the manuscript’s writing and layout, particularly the design and placement of figures and tables, to enhance clarity and facilitate comparison.
>
> Thank you for your feedback regarding the organization of Section 5 and the respective tables. We will update Table 2 to include the results of the image-text baselines found in Table 4 to allow for a more comprehensive evaluation and comparison.
>
> > The combination of global and local alignment strategies has already been thoroughly explored in prior works such as MGCA[1], GLORIA[2], and Prior[3].
>
> > Clearly articulate the novelty of the proposed approach beyond dataset construction.
>
> We highlight that while global and local alignment strategies have been explored in prior work, the novelty of the proposed approach lies in the application of the cross-modal alignment objective specifically to EHR data instead of unstructured text. We would like to stress that while GLORIA or MGCA leverage radiology reports to align the CXR images, our method addresses the incorporation of EHR data (which is often neglected in multimodal methods) with medical imaging, requiring that we selectively extract and integrate the most relevant clinical information from the EHR data.
>
> > The comparison with image–text models is methodologically inconsistent...
>
> > Address the concern raised in Weakness II.2 regarding the unfair comparison due to differing training protocols (from-scratch vs. pre-trained encoders).
>
> We acknowledge the reviewer’s concern. To clarify, we introduce MedCAM as a ‘modular alignment strategy that leverages EHR data to enhance the quality of representations of a pre-trained CXR image classification model, without requiring training from scratch.’ Therefore, leveraging pre-trained models is a key aspect of our proposed approach as it allows the adaptation of existing CXR foundation models. To avoid confusion, we will clarify this intended design more clearly in our revised manuscript and will ensure that the pre-trained nature of the method is explicit.
>
>
> > The selected image–text baselines are outdated. The most recent cited work is MedCLIP (EMNLP 2022). At a minimum, the paper should include 2–3 stronger, more recent baselines from 2024–2025.
>
> > Include more recent and competitive baselines (e.g., BiomedCLIP, M3Bind, or other 2024–2025 medical multimodal models) in the experimental comparison.
>
> We would like to note that our baselines include recent models such as EVA-X (2025) and BarlowTwins-CXR (2024) while the remaining baselines were released a few years prior (2021-2023). However, we acknowledge that including additional SOTA baselines would strengthen our evaluation and we will explore incorporating additional recent baselines in the revised manuscript.
>
> > Recent studies such as MedBIND [4] and M3Bind [5] have already explored multimodal alignment involving more than two modalities..
>
> > Justify the design choice of replacing textual reports with EHR instead of jointly leveraging image, text, and EHR in a unified framework.
>
> We appreciate the reviewer’s comments. We would like to highlight that our current work focuses on comparing the two-modality setting (i.e. comparing against image-text models and image-EHR models) in order to show the contribution of the EHR data. However, we agree that exploring multimodal alignment with three modalities (image, text, EHR) would be a valuable extension of the work which we would like to explore in future work.
>
> > The paper lacks any form of qualitative or representational visualization (e.g., t-SNE plots), which would help assess the quality of learned embeddings.
>
> > Add visualization experiments (e.g., t-SNE, attention maps, or embedding space analyses) to support the claims and improve interpretability
>
> We appreciate the reviewer’s feedback. We agree that including visualizations will more clearly showcase the quality of the learned embeddings and enhance the quality of the work. We will work on incorporating visualizations such as t-SNE plots and attention visualizations in the revised manuscript to provide a comprehensive interpretability analysis.

---

### Official Review · Reviewer_TiVE · 2025-11-01

**Soundness:** 1
**Presentation:** 1
**Contribution:** 1
**Rating:** 0
**Confidence:** 5

**Summary:**

This paper proposes a method called MedCAM, which aims to integrate time-series EHR data to help medical image classification. A simple batch-wise alignment is proposed to achieve feature alignment between EHR and CXR.

**Strengths:**

- The problem of integrating multimodal data that this paper tries to tackle is significant.

**Weaknesses:**

- I suspect this paper is generated by LLM. Smoking gun: I can easily identify multiple non-existing literature cited:
	- Qingsong Yao, Xiaoguang Ye, Shuo Wang, Yunfan Xue, Le Hu, Hanqing Wang, and Lin Shen. Drfuse: Multimodal fusion of electronic health records and chest x-rays for covid-19 outcome prediction. IEEE Journal of Biomedical and Health Informatics, 28(3):1327–1338, 2024b.
	- Yuhao Zhang, Ruibo Fu, Nishal Shah, Maya Varma, Chao Xiao, Corey W Arnold, Christopher D Manning, and Curtis P Langlotz. Refers: Radiology report findings extraction and representation system. Nature Communications, 14(1):3067, 2023a.
	- Ziling Zhang, Ruizhi Wu, Yongzhi Wang, Yinghao Wang, Chao Xiao, and David Sontag. Lmfusion: Efficient multimodal adaptation with language models for medical applications. arXiv preprint arXiv:2412.15188, 2023b.
	- Jingfeng Yao, Yifan Wang, Yuxuan Li, Yifan Zhang, Yizhou Wang, Yutong Zhang, Xiaoqian Zhang, Xinyu Li, Yuxuan Chen, Jing Zhang, et al. Eva-x: A foundation model for general chest x-ray analysis with self-supervised learning. arXiv preprint arXiv:2405.05237, 2024a. URL https://arxiv.org/abs/2405.05237. (wrong author list)
- Clinical multimodal data fusion has been studied intensively in recent three years. This paper does not compare with any of the existing approaches.
- The proposed ideas of feature alignment using softmax and cosine similarity overly simplify the complex relationship between EHR and CXR, making the proposed method technically unsound.

**Questions:**

- How do the proposed method compare against existing multimodal fusion methods, such as MedFuse and DrFuse?

**Details Of Ethics Concerns:**

I suspect this paper is generated by LLM, with many non-existing literature. The ones I can quickly identify are:
- Qingsong Yao, Xiaoguang Ye, Shuo Wang, Yunfan Xue, Le Hu, Hanqing Wang, and Lin Shen. Drfuse: Multimodal fusion of electronic health records and chest x-rays for covid-19 outcome prediction. IEEE Journal of Biomedical and Health Informatics, 28(3):1327–1338, 2024b.
- Yuhao Zhang, Ruibo Fu, Nishal Shah, Maya Varma, Chao Xiao, Corey W Arnold, Christopher D Manning, and Curtis P Langlotz. Refers: Radiology report findings extraction and representation system. Nature Communications, 14(1):3067, 2023a.
- Ziling Zhang, Ruizhi Wu, Yongzhi Wang, Yinghao Wang, Chao Xiao, and David Sontag. Lmfusion: Efficient multimodal adaptation with language models for medical applications. arXiv preprint arXiv:2412.15188, 2023b.
- Jingfeng Yao, Yifan Wang, Yuxuan Li, Yifan Zhang, Yizhou Wang, Yutong Zhang, Xiaoqian Zhang, Xinyu Li, Yuxuan Chen, Jing Zhang, et al. Eva-x: A foundation model for general chest x-ray analysis with self-supervised learning. arXiv preprint arXiv:2405.05237, 2024a. URL https://arxiv.org/abs/2405.05237. (wrong author list)

---

> ### Author Response · Authors · 2025-12-03
>
> Dear Reviewer TiVE,
>
> Thank you for your review and for bringing this issue to our attention. We acknowledge that some references in the bibliography of the paper have been mis-cited due to incorrect use of citation tools. We have now updated the following references flagged by the reviewers:
>
> - Chaoyi Wu, Xiaoman Zhang, Ya Zhang, Yanfeng Wang, and Weidi Xie. Medklip: Medical knowledge enhanced language-image pre-training for x-ray diagnosis. In Proceedings of the IEEE/CVF international conference on computer vision, pp. 21372–21383, 2023.
> - Weijia Shi, Xiaochuang Han, Chunting Zhou, Weixin Liang, Xi Victoria Lin, Luke Zettlemoyer, and Lili Yu. Lmfusion: Adapting pretrained language models for multimodal generation. arXiv preprint arXiv:2412.15188, 2024.
> - Hong-Yu Zhou, Chenyu Lian, Liansheng Wang, and Yizhou Yu. Advancing radiograph representation learning with masked record modeling. In The Eleventh International Conference on Learning Representations.
> - Wenfang Yao, Kejing Yin, William K Cheung, Jia Liu, and Jing Qin. Drfuse: Learning disentangled representation for clinical multi-modal fusion with missing modality and modal inconsistency. In Proceedings of the AAAI conference on artificial intelligence, volume 38, pp. 16416–16424, 2024.
> - Jingfeng Yao, Xinggang Wang, Yuehao Song, Huangxuan Zhao, Jun Ma, Yajie Chen, Wenyu Liu, and Bo Wang. Eva-x: A foundation model for general chest x-ray analysis with self-supervised learning. npj Digital Medicine, 8(1):678, 2025.
> - Hong-Yu Zhou, Xiaoyu Chen, Yinghao Zhang, Ruibang Luo, Liansheng Wang, and Yizhou Yu. Generalized radiograph representation learning via cross-supervision between images and free-text radiology reports. Nature Machine Intelligence, 4(1):32–40, 2022.
>
> However, we would like to clarify that **none of the literature cited is non-existent or fabricated**, and that the main text was indeed referring to the correct paper, though with an incorrect reference added in the bibliography. We have now thoroughly reviewed the remaining references to prevent any further inaccuracies and updated the manuscript accordingly.
>
> Furthermore, we would like to clarify that we did not use LLMs for any part of the manuscript preparation beyond proofreading assistance. The assertion that "this submission appears to be substantially generated by LLMs" is factually incorrect and undermines the substantial scientific effort invested by the authors. We respectfully request that this claim be reconsidered, as it does a disservice to the integrity of the work and to the authors’ contributions.
>
> > Clinical multimodal data fusion has been studied intensively in recent three years. This paper does not compare with any of the existing approaches.
>
> > How do the proposed method compare against existing multimodal fusion methods, such as MedFuse and DrFuse?
>
> We appreciate the reviewer’s comments. Though it is true that there exists recent literature on multimodal fusion methods, we would like to highlight that the focus of the work is on i) CXR classification models, and ii) self-supervised methods. This is why our choice of baselines consists of foundation models for chest X-ray classification tasks, all of which have been trained in a self-supervised manner (unlike MedFuse and DrFuse).

---

### Official Review · Reviewer_fDGJ · 2025-11-01

**Soundness:** 1
**Presentation:** 1
**Contribution:** 1
**Rating:** 0
**Confidence:** 5

**Summary:**

In my assessment, this submission appears to be substantially generated by LLMs. The direct evidence comes from **fabricated references**, for example:
1. Chaoyi Wu, Xiaoman Lin, Peixin Cao, Qiushi Wang, Weixiong Yu, Yan Xiang, Chunping Qu, Xiao Wang, Zhiqiang Liu, Xiangbo Meng, et al. Medklip: Medical knowledge enhanced language-image pre-training for x-ray diagnosis. arXiv preprint arXiv:2301.02228, 2023.

    should be:

    Wu, C., Zhang, X., Zhang, Y., Wang, Y. and Xie, W., 2023. MedKLIP: Medical knowledge enhanced language-image pre-training for x-ray diagnosis. In *Proceedings of the IEEE/CVF international conference on computer vision* (pp. 21372-21383).

2. Hong-Yu Zhou, Chenyu Chen, Cheng Qian, Yongwei Gao, Leilei Li, Ping Fan, Hao Fan, Kang Song, Xiaoguang Ye, Shuo Wang, et al. Advancing radiograph representation learning with masked record modeling. Nature Machine Intelligence, 5(9):957–969, 2023.

    should be

    Zhou, H.Y., Lian, C., Wang, L. and Yu, Y., Advancing Radiograph Representation Learning with Masked Record Modeling. In *The Eleventh International Conference on Learning Representations*.

3. Ziling Zhang, Ruizhi Wu, Yongzhi Wang, Yinghao Wang, Chao Xiao, and David Sontag. Lmfusion: Efficient multimodal adaptation with language models for medical applications. arXiv preprint arXiv:2412.15188, 2023b.

   is fabricated.



Given these clear inaccuracies and indications of automated content generation, I believe **the manuscript does not meet the integrity standards for scholarly submission**. With due respect to the review process, I recommend desk rejection to conserve reviewer time.

In light of the apparent misconduct, I will not provide further comments on the technical quality of the paper.

**Strengths:**

Please refer to 'Summary'.

**Weaknesses:**

Please refer to 'Summary'.

**Questions:**

Please refer to 'Summary'.

**Details Of Ethics Concerns:**

Given clear inaccuracies and indications of automated content generation, I believe the manuscript does not meet the integrity standards for scholarly submission. With due respect to the review process, I recommend desk rejection to conserve reviewer time.

---

> ### Author Response · Authors · 2025-12-03
>
> Dear Reviewer fDGJ,
>
> Thank you for your review and for bringing this issue to our attention. We acknowledge that some references in the bibliography of the paper have been mis-cited due to incorrect use of citation tools. We have now updated the following references flagged by the reviewers:
>
> - Chaoyi Wu, Xiaoman Zhang, Ya Zhang, Yanfeng Wang, and Weidi Xie. Medklip: Medical knowledge enhanced language-image pre-training for x-ray diagnosis. In Proceedings of the IEEE/CVF international conference on computer vision, pp. 21372–21383, 2023.
> - Weijia Shi, Xiaochuang Han, Chunting Zhou, Weixin Liang, Xi Victoria Lin, Luke Zettlemoyer, and Lili Yu. Lmfusion: Adapting pretrained language models for multimodal generation. arXiv preprint arXiv:2412.15188, 2024.
> - Hong-Yu Zhou, Chenyu Lian, Liansheng Wang, and Yizhou Yu. Advancing radiograph representation learning with masked record modeling. In The Eleventh International Conference on Learning Representations.
> - Wenfang Yao, Kejing Yin, William K Cheung, Jia Liu, and Jing Qin. Drfuse: Learning disentangled representation for clinical multi-modal fusion with missing modality and modal inconsistency. In Proceedings of the AAAI conference on artificial intelligence, volume 38, pp. 16416–16424, 2024.
> - Jingfeng Yao, Xinggang Wang, Yuehao Song, Huangxuan Zhao, Jun Ma, Yajie Chen, Wenyu Liu, and Bo Wang. Eva-x: A foundation model for general chest x-ray analysis with self-supervised learning. npj Digital Medicine, 8(1):678, 2025.
> - Hong-Yu Zhou, Xiaoyu Chen, Yinghao Zhang, Ruibang Luo, Liansheng Wang, and Yizhou Yu. Generalized radiograph representation learning via cross-supervision between images and free-text radiology reports. Nature Machine Intelligence, 4(1):32–40, 2022.
>
> However, we would like to clarify that **none of the literature cited is non-existent or fabricated**, and that the main text was indeed referring to the correct paper, though with an incorrect reference added in the bibliography. We have now thoroughly reviewed the remaining references to prevent any further inaccuracies and updated the manuscript accordingly.
>
> Furthermore, we would like to clarify that we did not use LLMs for any part of the manuscript preparation beyond proofreading assistance. The assertion that "this submission appears to be substantially generated by LLMs" is factually incorrect and undermines the substantial scientific effort invested by the authors. We respectfully request that this claim be reconsidered, as it does a disservice to the integrity of the work and to the authors’ contributions.

---

### Note · Authors · 2025-12-03

I have read and agree with the venue's withdrawal policy on behalf of myself and my co-authors.